# Truthfulness Does Not Scale Like Reasoning: Why Polling Fails as a Proxy Verifier

Yegor Denisov-Blanch [1] [*]   Joshua Kazdan [2] [*]   Jessica Chudnovsky [1]   Rylan Schaeffer [1]   Sheng Guan [3]
Soji Adeshina [3]   Sanmi Koyejo [1]

## Abstract

Pass@$k$ and other methods of scaling inference compute can improve language model performance in domains with external verifiers, including mathematics and code, where incorrect candidates can be filtered reliably. This raises a natural question: can we similarly scale compute to elicit gains in *truthfulness* for domains without convenient verification? We show that across five benchmarks and models, surprisingly, it cannot. Even at $25\times$ the inference cost of naive sampling, polling-style aggregation yields no consistent accuracy gains over single-sample baselines and often amplifies correlated errors. We find that under uncertainty, models are better at predicting what other models will say within model ensembles than at identifying what is true, revealing a separation between *social prediction* and *truth verification*. Across models and benchmarks, aggregation fails to provide a robust truth signal because language model errors are strongly correlated. The source of correlation goes beyond any individual benchmark: we show that even when conditioned on out of distribution random strings and asked to produce pseudo-random outputs, different models produce correlated outputs. Confidence-based weighting provides no benefit because self-reported confidence fails to reliably distinguish correct from incorrect answers. These results delineate a boundary for inference-time scaling: in verified domains, additional samples provide more candidates for a verifier to filter; in unverified domains, additional samples merely reinforce correlated errors.

[*]Equal contribution  [1]Stanford Computer Science [2]Stanford Statistics [3]Amazon. Correspondence to: Yegor Denisov-Blanch <ydebl@stanford.edu>, Joshua Kazdan <jkazdan@stanford.edu>, Sanmi Koyejo <sanmi@cs.stanford.edu>.

*Proceedings of the 43$^{rd}$ International Conference on Machine Learning*, Seoul, South Korea. PMLR 306, 2026. Copyright 2026 by the author(s).

## 1. Introduction

Inference-time scaling has emerged as a powerful alternative to parameter scaling for large language models. By allocating additional compute at test time, models can generate multiple candidate solutions and select among them, often achieving accuracy gains. Methods such as self-consistency (Wang et al., 2023), repeated sampling (Brown et al., 2024), and search have proven effective in domains like mathematics and code, where correctness can be verified automatically. In these settings, aggregation amplifies capability by filtering candidates through an external verifier.

A natural question is whether aggregation can also scale *truthfulness* in domains without verification. Selection must then instead rely on internal signals such as agreement, confidence, or predicted popularity. A common intuition, borrowed from the wisdom-of-crowds literature (Surowiecki, 2004), is that aggregating many imperfect judgments should recover the truth even when individuals err.

We find that this intuition does not transfer to language models. We evaluate multiple aggregation strategies—including majority voting, confidence weighting, and the Surprisingly Popular algorithm (Prelec et al., 2017)—across several open-source models and benchmarks spanning factual knowledge, commonsense reasoning, expert-level questions, and forecasting. We find that no aggregation method consistently improves accuracy over single-sample baselines; methods that help on one benchmark often hurt on another. On forecasting questions with outcomes postdating model knowledge cutoffs, all methods perform at chance. These results do not contradict prior successes of self-consistency; they identify the regime in which those methods stop working.

This failure traces to correlated errors. Wisdom-of-crowds rests on a critical assumption: errors must be at most weakly correlated. Human crowds satisfy this condition because individuals draw on diverse experiences and information sources. Modern language models do not. Models trained on overlapping corpora and optimized for similar objectives produce correlated errors and shared failure modes. This parallels adversarial transferability, where different models fail in similar ways because they learn similar features

(Goodfellow et al., 2015). When one model produces a plausible but incorrect answer, others frequently do the same. Polling does not cancel these mistakes; it amplifies correlated errors, producing greater consensus without greater correctness.

What is the source of these correlations? One might suspect they stem entirely from shared knowledge and misconceptions encoded in overlapping training data. To test this, we feed models randomly generated ASCII strings and ask for a forced multiple-choice answer—a setting with no ground-truth signal whatsoever. Even on this zero-knowledge baseline, different LMs exhibit correlations as high as 0.35. This rules out shared factual knowledge as the sole explanation and suggests that correlated outputs arise from aligned inductive biases and architectural similarities, therefore persist regardless of aggregation.

We diagnose these failures mechanically. Model errors are highly correlated both within and across model families, violating the independence that crowd wisdom requires. Self-reported confidence is misaligned with correctness and instead tracks expected agreement. Surprise-based signals that power sophisticated aggregation rules do not reliably distinguish truth from consensus. At the same time, models predict collective opinion substantially better than they predict correctness, revealing a separation between *social prediction* and *truth verification*.

Together, these results define a boundary for inference-time scaling. Aggregation improves performance when a verifier exists, but it cannot serve as a verifier substitute. Scaling truthfulness in unverifiable domains requires external grounding or interventions that break error correlation, rather than instantiation of the bitter lesson (Sutton, 2019) involving more samples from the same shared generative distribution.

To summarize our conclusions:

1. We show that polling-style aggregation fails to reliably improve truthfulness across verifier-absent benchmarks even at large inference-time compute.

2. We demonstrate that language model errors are strongly correlated across samples and across model families.

3. We explain why aggregation methods like confidence, predicted popularity, and surprise reflect consensus rather than correctness.

4. We introduce a simple negative control to probe the depth of inter-model correlations - random strings with forced-choice answers - in which no underlying truth exists, and show persistent above-chance agreement. This provides mechanistic evidence that correlation reflects shared inductive biases rather than shared knowledge.

## 2. Related Work

Inference-time scaling complements parameter scaling by allocating compute at test time. Self-consistency decoding samples multiple reasoning paths and selects answers by majority vote (Wang et al., 2023). Subsequent work showed log-linear gains with additional samples (Brown et al., 2024; Hughes et al., 2025), explained these gains via heavy-tailed difficulty distributions (Schaeffer et al., 2025), and demonstrated that adaptive test-time compute can outperform fixed best-of-$N$ strategies (Snell et al., 2025). A parallel line of work studies sample-and-aggregate systems directly. Li et al. (2024) find that performance can improve by instantiating more agents and aggregating their answers, while Chen et al. (2024) analyze Vote and Filter-Vote systems and show that performance can saturate or even decline as the number of model calls increases. Chen et al. (2025) similarly identify saturation in test-time aggregation and improve reasoning by adding iterative refinement, external reward-model scores, and multi-agent feedback. Together, these results show that sample-and-aggregate methods can be useful, while also identifying saturation and failure modes of naive voting. Our work asks a different question: whether aggregation can function as a proxy verifier for truthfulness when no external correctness signal is available.

In domains without external verifiers, selection must rely on internal signals. One proposal is to treat aggregation itself as a proxy for verification, drawing on the wisdom-of-crowds literature (Surowiecki, 2004). Averaging independent judgments often outperforms any individual, and repeated estimates from a single person can reduce variance through an "inner crowd" effect (van Dolder & van den Assem, 2018). Repeated samples from a language model form an inner crowd, while pooling across models forms an outer crowd, analogous to ensembles in machine learning.

Classical ensemble theory predicts gains when component errors are diverse or weakly correlated (Lakshminarayanan et al., 2017). In binary tasks, this assumption concerns *error events*, not raw agreement: errors are correlated if, conditioned on being wrong, models tend to select the *same* incorrect option. Modern language models increasingly violate this condition. When different LLMs err on the same question, they often collapse onto a single wrong answer (Kim et al., 2025), and error correlation grows with model capability and training similarity (Goel et al., 2025). As a result, aggregation may amplify correlated errors rather than cancel noise.

Several aggregation methods aim to recover truth when the majority is wrong. The Surprisingly Popular (SP) algorithm compares vote shares to predicted ones, exploiting asymmetries between informed and uninformed respondents (Prelec et al., 2017). For SP to succeed, some respondents must know the truth and anticipate the majority's error. While

SP-style aggregation has been explored for language models, it remains unclear whether LLM populations exhibit this expert-minority structure. We test both prerequisites – knowing the truth and anticipating majority error – directly.

Other aggregation schemes weight or score samples using model-internal confidence signals, assuming that higher-confidence answers are more likely to be correct. Soft self-consistency replaces exact majority voting with a continuous likelihood-based score, improving efficiency when exact answer matches are sparse (Wang et al., 2024). Confidence-informed self-consistency instead weights answers by self-reported confidence and improves sample efficiency on reasoning benchmarks (Taubenfeld et al., 2025). These methods study whether confidence can improve selection among multiple samples from a model. We study a different regime: whether confidence, predicted popularity, and surprise can serve as endogenous proxy verifiers across model ensembles in verifier-absent truthfulness tasks. Although base language models can be calibrated (Kadavath et al., 2022), post-training often degrades this property (Tian et al., 2023; Leng et al., 2025; Xiong et al., 2024). Sycophancy further encourages agreement over truth (Sharma et al., 2023), suggesting that confidence may track expected consensus rather than epistemic certainty.

Relatedly, work on self-correction shows that absent external feedback, models often rationalize errors rather than detect them (Huang et al., 2024). Since crowd aggregation relies on partially independent errors, shared training data and objectives pose a fundamental obstacle.

Prior work shows that inference-time compute helps scale performance when verifiers exist, and that LLM errors are often correlated. What remains unresolved is whether this correlation limits *truthfulness* in verifier-absent domains. We show that it does: correlated errors impose a structural limit that self-aggregation rules don't overcome.

## 3. Experimental Setup

Crowd wisdom relies on errors that are not strongly correlated. Our experiments test this assumption by applying polling methods to repeated samples from single models (*intra-model crowds*) and to ensembles pooled across models (*inter-model crowds*), while measuring whether the dependence structure required for aggregation is satisfied.

**Tasks and response formats.** We elicit three response types: (i) a binary answer (e.g., YES/NO or TRUE/FALSE), (ii) an answer with self-reported confidence on a 0–100 scale, and (iii) a prediction of the vote share of one canonical option (e.g., the fraction of responses answering YES or TRUE). We use the 0–100 confidence scale as a black-box elicitation format, not as a claim that models provide

reliable or calibrated probabilities at this granularity; future work should evaluate coarser numerical scales, Likert-style ratings, and human-labeled uncertainty judgments. Aggregation rules such as majority voting and the Surprisingly Popular (SP) algorithm require a finite option space with well-defined vote shares and prediction distributions. Binary questions are therefore not a requirement but the simplest setting, avoiding discretization artifacts and enabling comparison across aggregation methods. This choice limits generalization to open-ended generation, which we discuss in Section 5. Binary questions represent the most favorable setting for polling-style aggregation; failure to obtain reliable gains even in this simplified regime provides strong evidence against aggregation as a general mechanism for scaling truthfulness in verifier-absent domains.

We evaluate four benchmarks spanning verifier-absent regimes:

- **Com2Sense** (Singh et al., 2021): binary commonsense reasoning.

- **Humanity's Last Exam (HLE)** (Phan et al., 2025): expert-level questions, restricted to those with a naturally binary answer structure.

- **BoolQ** (Clark et al., 2019): binary factual question answering.

- **Predict-the-Future**: a collection of forecasting questions introduced in this work, where all ground-truth outcomes were *definitively resolved and verifiable at the time of dataset construction and writing*, even though the events postdate model knowledge cutoffs.

The HLE subset yields wider confidence intervals due to its small size, but accuracy remains far below the 50% guessing baseline (5.7%), indicating systematic attraction to incorrect answers rather than sampling noise.

**Sampling protocol.** For each question and response type, we collect 25 independent samples per model at temperatures $T \in \{0.7, 1.0\}$. For models that support chain-of-thought reasoning, reasoning is enabled during generation; our evaluation therefore aggregates final answers produced after model reasoning rather than direct-answer-only completions. We use moderate-to-high temperatures because crowd aggregation requires diversity; at low temperatures, repeated sampling collapses to near-deterministic outputs. The relevant question is not whether a single greedy sample outperforms high-temperature aggregation, but whether aggregation provides gains *given* the diversity required for crowd methods to apply. Across all benchmarks, this corresponds to 50 samples per (question, model) and a total of *375,000* model responses across experiments; we report benchmark-wise counts in Appendix B.

**Models and crowds.** We evaluate five instruction-tuned, open-source models spanning 4B to 235B parameters across three families: Gemma-3-4B (Gemma Team, 2025), GPT-oss-20B and GPT-oss-120B (Agarwal et al., 2025), Qwen-32B (Yang et al., 2024), and Qwen3-235B (Qwen Team, 2025). Intra-model crowds consist of repeated samples from a single model. Inter-model crowds pool responses across all five models at $T = 1.0$, yielding 125 votes per question. Evaluating both regimes allows us to test whether increased architectural and training diversity reduces error correlation, as crowd wisdom would predict.

**Aggregation methods.** We evaluate five aggregation rules that exhaust common internal selection signals: (1) Majority Vote (de Condorcet, 1785); (2) Highest Confidence (Kadavath et al., 2022); (3) Confidence-Weighted Vote (Wang et al., 2023; Taubenfeld et al., 2025); (4) Prediction-Weighted Vote, weighting each response by its predicted popularity; and (5) Surprisingly Popular (SP) (Prelec et al., 2017), which selects the answer whose observed support exceeds its predicted support. If self-aggregation can scale truthfulness without an external verifier, it should do so under at least one of these rules. We additionally report an inverse-SP diagnostic solely to test alignment between the SP signal and correctness; it is not proposed as a method.

**Evaluation and correlation.** Performance is measured by accuracy with 95% bootstrap confidence intervals obtained by resampling questions. We use these question-level bootstrap intervals as our primary significance procedure, so all reported uncertainty reflects variation over evaluation questions rather than over samples from the same question. Because all methods are evaluated on the same questions, comparisons between aggregation methods are interpreted primarily through question-level paired differences and bootstrap uncertainty rather than as independent accuracy estimates. Because independence is the key assumption under test, correlation is a first-class quantity. We measure it via majority stability under temperature variation, inter-rater reliability (including vote entropy and Fleiss' $\kappa$), and the concentration of wrong answers when models err. These measures distinguish genuine epistemic diversity from surface-level variation.

**Controls and forecasting benchmark.** To distinguish structural correlation from shared knowledge, we include control analyses in which no correct answer exists, allowing us to test whether model agreement persists even when no signal is present. Finally, we include Predict-the-Future, a forecasting evaluation set whose outcomes postdate model knowledge cutoffs. All outcomes are manually verified. If aggregation can extract truth from uncertain respondents, it should help in this setting; accuracy near chance therefore provides a stringent negative test.

# 4. Results

Across verifier-absent benchmarks, we find that self-aggregation does not provide a reliable path from inference-time compute to truthfulness. Despite substantial increases in consensus, accuracy remains flat or degrades. This pattern holds even though reasoning is enabled for models that support it, so the negative result is not an artifact of direct-answer-only prompting. This failure is systematic rather than accidental. We show that when language models err, their errors are often correlated, violating the independence assumptions required for crowd wisdom. The same correlation structure appears in both majority-correct and majority-incorrect cases, indicating that agreement is not selectively informative about truth. Consequently, when model errors are correlated, aggregation rules based on internal signals like agreement, confidence, or predicted popularity cannot reliably scale truthfulness.

## 4.1. Aggregation Fails to Improve Truthfulness

We evaluate five aggregation rules (majority vote, highest confidence, confidence-weighted vote, prediction-weighted vote, and Surprisingly Popular) across four verifier-absent benchmarks: Humanity's Last Exam (HLE), BoolQ, Com2Sense, and Predict-the-Future. Across models, temperatures, and datasets, no method consistently outperforms single-sample baselines (Figure 2). Increasing the number of samples increases consensus, but not correctness.

The forecasting benchmark provides the clearest negative test: because outcomes postdate model knowledge cutoffs, aggregation should succeed here if it could extract latent expertise. Instead, all methods remain indistinguishable from chance.

These failures are structural. Crowd wisdom requires errors to be uncorrelated enough that mistakes cancel under aggregation (Surowiecki, 2004; de Condorcet, 1785). Language models violate this assumption - shared training data, objectives, and post-training incentives produce correlated errors and shared failure modes. Hence, aggregation increases consensus without increasing correctness.

Since majority voting fails, we next examine whether other model-internal signals - like confidence or predicted popularity - can recover correctness. We test confidence-based weighting and the Surprisingly Popular (SP) algorithm. Both fail because, in practice, they end up measuring expected consensus rather than truth.

Confidence does not track correctness. Self-reported confidence correlates weakly with accuracy but similarly with agreement. On hard benchmarks, confident answers are often wrong - due to sycophantic training that rewards typical-sounding outputs (Sharma et al., 2023; Leng et al., 2025).

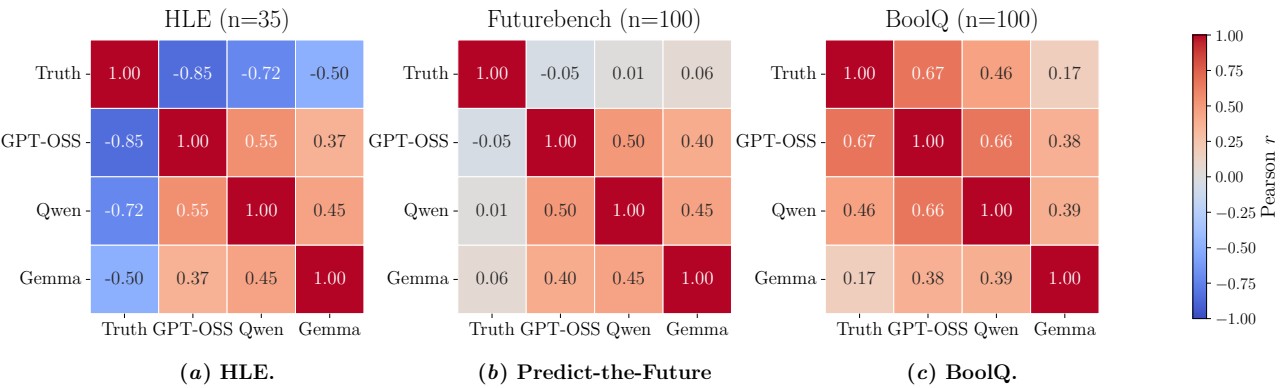

*Figure 1.* **Models agree with each other more reliably than they agree with truth.** Each panel reports Pearson correlations between binary answers from different model families and the ground-truth label ("Truth"), computed over questions in the benchmark. Across datasets, inter-model correlations are consistently positive, but correlation with Truth varies sharply by task, showing that agreement is not a stable proxy for correctness.

The SP algorithm assumes an expert minority that both knows the truth and anticipates the majority's error (Prelec et al., 2017). Language model populations rarely exhibit this structure. As a diagnostic, we evaluate inverse-SP. On HLE, inverse-SP attains 80% accuracy, implying that the standard SP signal is systematically anti-correlated with correctness. On other datasets, inverse-SP performs at chance. The sign of the surprise gap is therefore not stable across tasks, meaning it cannot serve as a reliable verification signal.

### 4.2. Correlated Errors Explain the Failure

Polling-style aggregation relies on sufficiently diverse and independent errors, under which aggregation increases the probability of correctness (Surowiecki, 2004; de Condorcet, 1785). We test this assumption directly by measuring cross-family answer correlations and their alignment with ground truth (Figure 1). We also repeat the correlation analysis after segregating questions by whether the ensemble majority is correct or incorrect; the resulting correlation structure is consistent across both subsets, suggesting that correlations are not a usable signal for distinguishing truth amplification from false-signal amplification.

**Correlated mistakes in verifiable domains.** We analyze error correlation in verifiable mathematics benchmarks - MATH (Hendrycks et al., 2021) and AIME (Mathematical Association of America, 1983–present) - where aggregation methods are known to help due to the presence of external verifiers. For each dataset, we evaluate 128 problems with 200 samples per problem and focus on *plurality-wrong* cases, where the most frequent answer is incorrect.

On MATH, plurality-wrong rates are low (3.9–25.8%), but errors are highly concentrated: the most common incorrect answer accounts for 65–87% of wrong responses. On AIME,

plurality-wrong rates vary widely (3.9–75.8%) and error concentration is substantially lower (20–63%).

These results clarify the distinction between two forms of inference-time scaling. In pass@$k$ and best-of-$k$ settings with external verifiers, additional samples help because incorrect candidates can be filtered by proof checkers, unit tests, or other correctness checks. Self-consistency proper can also improve mathematics without an explicit verifier by marginalizing over sampled reasoning paths. Our point is narrower: in verifier-absent truthfulness tasks, agreement alone is not a reliable correctness signal. When models converge on the same wrong answer, polling has no external check that can distinguish shared error from shared truth.

**Surprise signals do not reliably track truth.** The Surprisingly Popular (SP) algorithm succeeds only when an expert minority both knows the truth and anticipates the majority's error. Using inverse-SP purely as a diagnostic, we find that the direction of the surprise gap is not stable across benchmarks. On HLE, inverse-SP attains high accuracy, implying that the standard SP signal is systematically anti-correlated with correctness. On other datasets, inverse-SP performs at or near chance.

The surprise gap points toward truth on some benchmarks and away from it on others. A signal that flips direction across tasks cannot serve as a verifier.

**Temperature sampling does not induce independence.** Varying temperature has little effect on inducing diversity: between $T=0.7$ and $T=1.0$, the *plurality* answer flips in only **2.9%** of (question, model) pairs (Table 2), indicating that additional samples explore surface variation rather than distinct hypotheses.

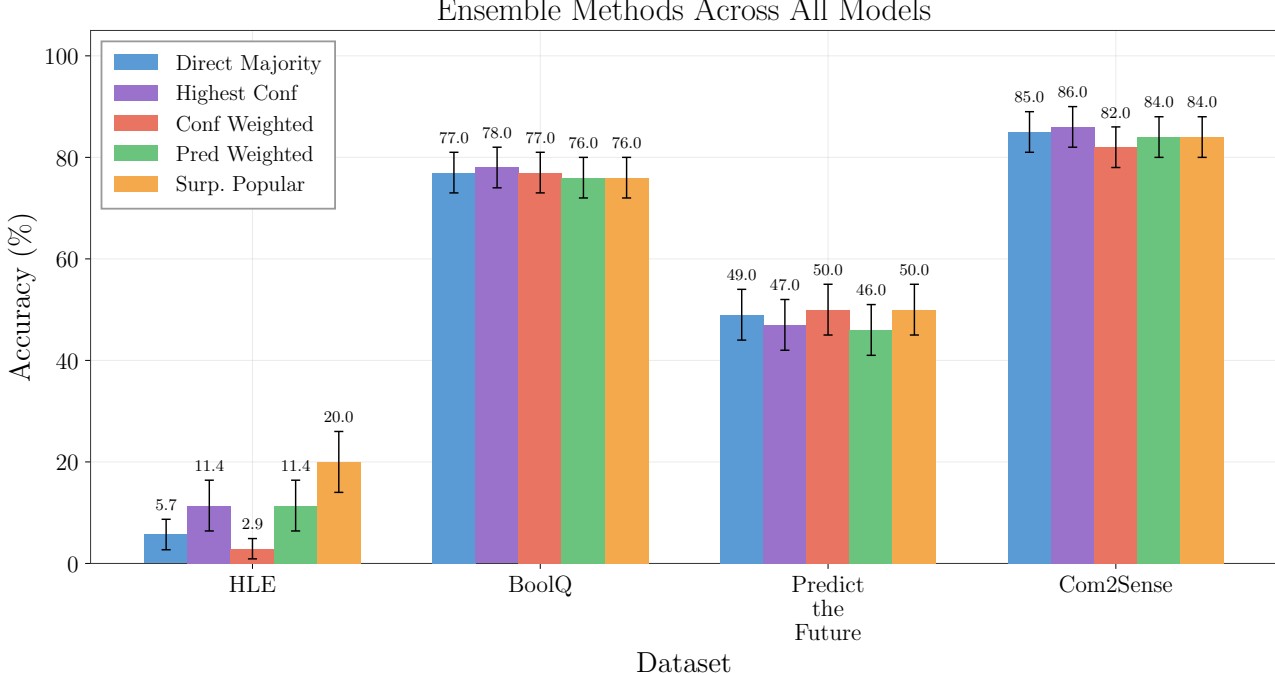

*Figure 2.* **No ensemble aggregation method consistently outperforms majority voting across benchmarks.** We compare five aggregation rules across four benchmarks using five-model ensembles (125 votes per question). Some methods improve performance on individual tasks, but none dominates overall. Error bars show 95% bootstrap confidence intervals.

Model ensembling likewise fails to restore independence. Despite large differences in overall error rates across models on both MATH and AIME, incorrect answers remain highly concentrated, and on 53% of MATH questions where multiple models err, they converge to the *same* incorrect answer.

Correlation is structural, not an artifact of temperature or model choice.

**Structural corollary.** When errors are correlated, no aggregation rule based solely on internal signals can reliably distinguish a unanimous correct answer from a unanimous wrong one. Additional samples increase confidence without increasing correctness.

### 4.3. Correlation Without Truth: A No-Signal Negative Control

Correlation on benchmark questions could reflect shared knowledge rather than structural dependence between models. To disentangle these, we introduce a negative control. We generate 10,000 prompts, each consisting of a uniformly random sequence of 32 characters, with models forced to choose from $\{A, B, C, D\}$:

> **Random String Prompt**
>
> ```
> Here is a random sequence:
>
> gP%!mdq4k!'q=T/rp~j~LdW05[:Mkdk$
>
> Now choose one option:  (A), (B), (C),
> or (D). Output your answer as X where
> X is A, B, C, or D.
> ```

Despite the absence of signal, model responses are not independent.

Across temperatures, several model pairs have a non-negligible positive correlation. Moreover, the qualitative structure of inter-model correlation is stable across $T=0.0$ and $T=1.0$, indicating that temperature increases surface diversity without eliminating shared response biases.

Models exhibit correlated outputs even when no truth exists, confirming that correlation stems from shared inductive biases in model weights, not shared knowledge. Polling then amplifies these biases, increasing consensus without improving correctness.

### 4.4. Consensus Is Easier Than Truth Verification

Models predict collective opinion far better than they predict correctness. Across benchmarks, vote-share predictions

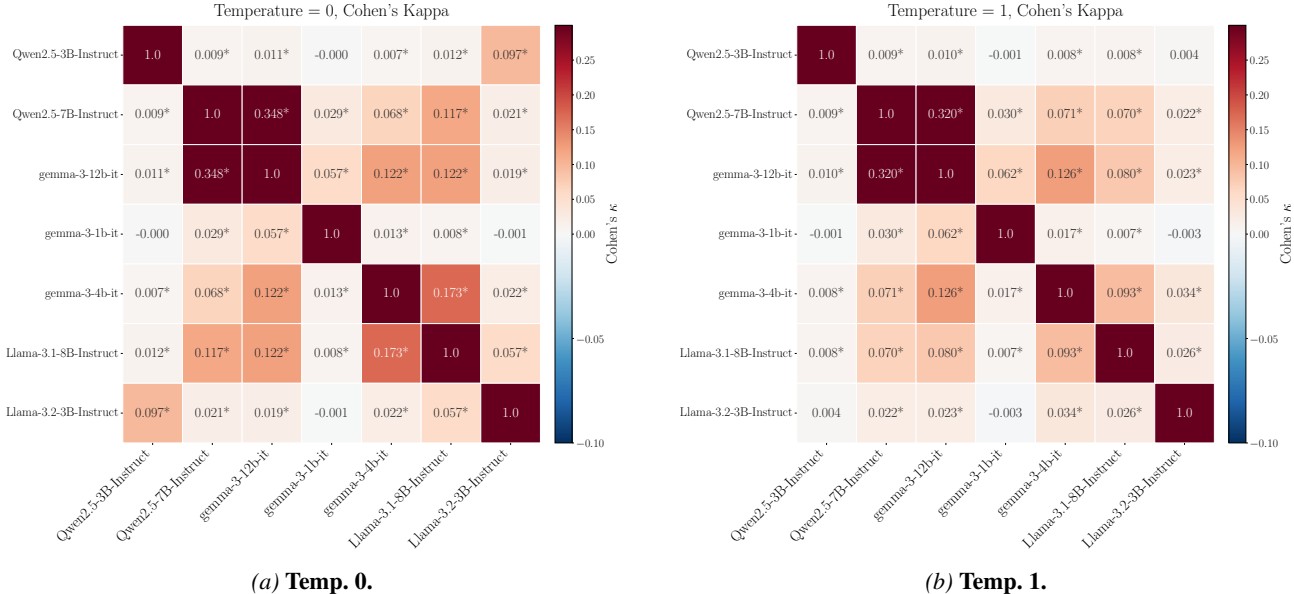

*(a)* **Temp. 0.**  *(b)* **Temp. 1.**

*Figure 3.* **Models exhibit correlated behavior even when no ground truth exists.** Cohen's $\kappa$ (Cohen, 1960) between pairs of models on random strings with forced-choice answers shows stable above-chance agreement, with a similar correlation structure across temperatures, indicating shared inductive biases rather than shared knowledge.

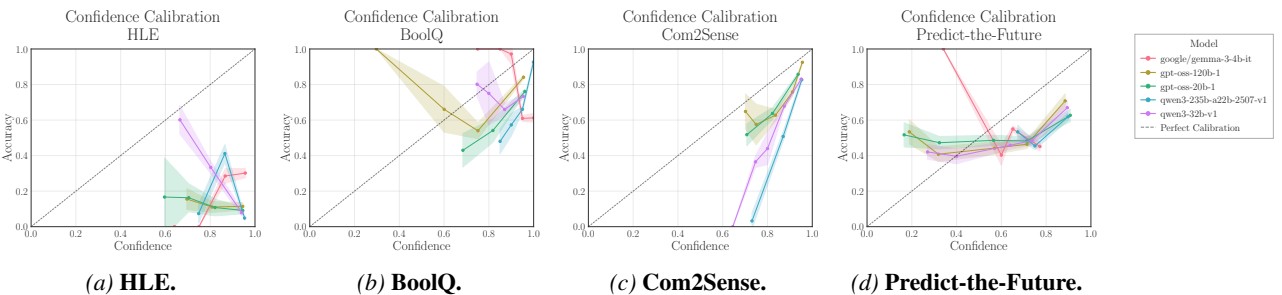

*(a)* **HLE.**  *(b)* **BoolQ.**  *(c)* **Com2Sense.**  *(d)* **Predict-the-Future.**

*Figure 4.* **Self-reported confidence is poorly aligned with correctness.** Reliability diagrams across four verifier-absent benchmarks. Confidence increases faster than accuracy and does not reliably distinguish correct from incorrect answers.

correlate strongly with vote shares, while confidence correlates weakly with accuracy. This asymmetry reveals that confidence and predicted popularity track what the crowd will say, not whether the crowd is right.

## 5. Discussion and Limitations

Our results identify a boundary condition for inference-time scaling. Sampling and aggregation improve performance when candidates can be externally verified. When no verifier exists, additional samples are typically drawn from the same epistemic prior. In this regime, aggregation increases consensus faster than it increases truth.

### 5.1. Why Internal Aggregation Signals Fail

Richer internal signals do not resolve this limitation. Agreement, confidence, predicted popularity, and surprise gaps

are all strong predictors of what the crowd will say. None provides a stable signal of correctness across tasks.

The Surprisingly Popular (SP) algorithm illustrates the problem. Its success depends on a stable expert-minority structure. Our diagnostics show that this structure is not reliably present in language model populations. The sign of the surprise gap varies across datasets, and in some cases is anti-correlated with correctness. A signal whose semantics change across tasks cannot function as a verifier.

Confidence-based methods fail for the same reason. Verbalized confidence tracks expected agreement nearly as strongly as it tracks accuracy. Weighting by confidence therefore amplifies the dominant error rather than correcting it.

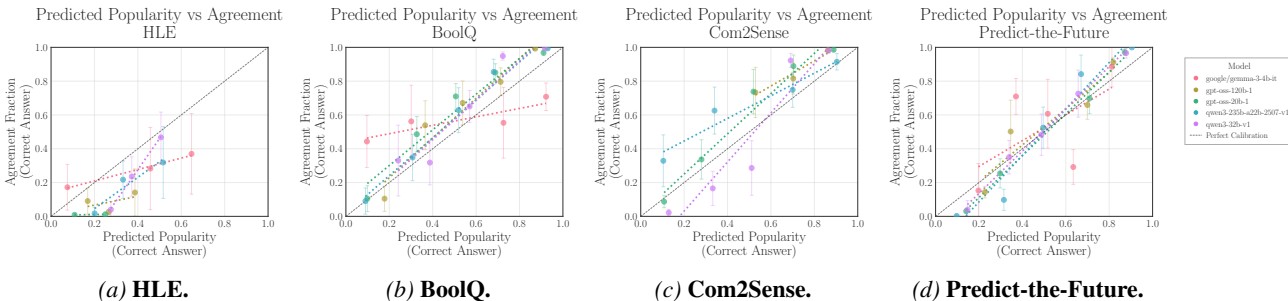

*Figure 5.* **Predicted popularity tracks agreement, not truth.** Predicted vote share for the correct option (x-axis) versus observed agreement (y-axis). Points denote models; error bars show 95% bootstrap confidence intervals. The dashed line indicates perfect calibration ($y = x$). Models accurately predict consensus even when consensus does not reliably indicate correctness.

## 5.2. Social Prediction Is Easier Than Truth Verification

Models predict collective opinion substantially better than they predict correctness. This reveals a separation between two behaviors that are easily conflated: *prediction consensus* (what will others say?) and *identifying correctness* (is this answer correct?). Most aggregation rules operate on the former. When the crowd is systematically wrong, these signals become actively misleading.

This distinction suggests a constructive use of social prediction.[1] Rather than selecting answers, predicted consensus can be used to flag high-risk questions and trigger retrieval, tool use, or deferral. In this framing, agreement is not evidence of truth, but a warning sign of shared failure modes. Thus, consensus may be useful as a control signal for routing questions to retrieval, tool use, abstention, or human review, but should not be used as a truth-selection mechanism in verifier-absent settings.

## 5.3. Limitations

**Binary response format.** We restrict to binary questions to make vote shares and surprise-based aggregation well-defined. This enables clean comparison across methods but limits direct applicability to open-ended generation. Extensions to open-ended settings typically introduce implicit verification steps, which do not contradict our core claim.

**Confidence elicitation.** We use verbalized confidence on a 0–100 scale to match common black-box usage, but we do not assume that models can reliably express calibrated probabilities at this granularity. The scale may be too fine-

grained and may invite arbitrary precision. While alternative uncertainty estimates may behave differently, the central failure mode is not merely noise but misalignment: confidence tracks consensus nearly as strongly as correctness. Future work should evaluate coarser numerical scales, Likert-style ratings, and human-labeled uncertainty judgments as alternative elicitation targets.

**Diversity interventions.** Temperature sampling and model ensembling change surface form more than the underlying response distribution. Stronger interventions that induce genuine independence typically introduce new information, objectives, or supervision. These effectively add verification or break shared priors, aligning with rather than refuting our conclusion.

**Forecasting benchmark scope.** Predict-the-Future is intentionally stringent but relatively small. Larger-scale replications would strengthen external validity.

**Reasoning traces.** Our experiments evaluate answer-level aggregation after model reasoning, not step-by-step correlations within reasoning traces. For models that expose or use reasoning, our results capture the final-answer effect of sampling over reasoning paths, but they do not identify whether shared errors arise early in the reasoning process or through convergence to the same final wrong answer. Understanding process-level error correlation in long-form reasoning is an important direction for future work.

## 5.4. What Would Change the Conclusion?

Our results apply to *self-aggregation without verification*. Truthfulness may scale when systems:

1. add grounding that functions as a verifier (retrieval, tools, execution, or human feedback),

2. engineer genuine epistemic diversity through disjoint training or objectives, or

---

[1]Throughout, we use "social prediction" and "truth verification" as labels for measured behavioral asymmetries: predicted vote share tracks observed vote share, while confidence tracks agreement more reliably than correctness. We do not mean to claim that these correspond to distinct internal faculties. Consistent with Section 4.3, we attribute the observed correlations to shared inductive biases in model weights rather than to shared beliefs or misconceptions.

3. learn explicit verifiers trained on externally labeled evidence.

All three introduce information absent from pure polling. Scaling truthfulness therefore requires verification or true independence, not more samples from a shared prior.

## 6. Conclusion

When language model errors are correlated, no aggregation rule based solely on agreement, confidence, or predicted popularity can reliably scale truthfulness without an external verifier.

Polling is not a substitute for verification. Across five instruction-tuned language models and multiple verifier-absent benchmarks, we find that agreement-based and metacognitive aggregation rules do not reliably improve accuracy over strong single-sample baselines, even at up to $25\times$ inference cost.

The failure is structural. Crowd wisdom relies on partially independent errors so that mistakes cancel under aggregation. Modern language models violate this assumption. Shared training data, objectives, and post-training incentives yield strongly correlated errors and shared failure modes. When the dominant answer is wrong, additional samples and additional models mostly reproduce the same mistake, turning aggregation into an amplifier of correlated errors rather than a mechanism for correction.

This yields a clear corollary. *When errors are correlated, no aggregation rule based solely on internal signals can reliably scale truthfulness in the absence of an external verifier.* Confidence, predicted popularity, and surprise gaps primarily track expected consensus, not correctness. Our diagnostics reveal a consistent separation between *social prediction* and *truth verification*: models are substantially better at forecasting what others will say than at identifying what is true. Signals that optimize the former cannot be repurposed into a verifier for the latter.

These results delineate a boundary for inference-time scaling. Aggregation improves performance when a verifier exists, because verification converts additional samples into additional correctness. In verifier-absent regimes, scaling truthfulness requires external grounding or interventions that actively break error correlation, rather than drawing more samples from a shared generative distribution. In short, inference-time compute scales reasoning when verification is available, but it does not scale truth itself.

Our results have important implications for the future of language models. Many problems, from reasoning to knowledge, have been solved simply by scaling up parameters, compute, and training data. However, as models become more capable, the importance of eliciting truthfulness and

obtaining reliable verifiers for scalable oversight will increase. Our results imply that the naive approach of throwing compute at the problem is unlikely to suffice. More creative solutions will be necessary to certify reliability and reinforce safety.

## Impact Statement

We show that having multiple language models vote on answers does not reliably improve accuracy in domains that lack a verifier. Since models often make the same mistakes, agreement can be misleading. We hope this encourages more realistic expectations about what aggregation methods can achieve, and pushes toward approaches that actually verify correctness rather than simply measure consensus.

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

# A. The Surprisingly Popular Algorithm

The Surprisingly Popular (SP) algorithm (Prelec et al., 2017) addresses a fundamental limitation of democratic aggregation: majority voting fails when most respondents lack the relevant knowledge. Standard approaches cannot distinguish between confident ignorance and genuine expertise. SP solves this by eliciting predictions about others' responses, exploiting an information asymmetry between those who know and those who do not.

**Setup.** Consider a binary question with answers $A$ and $B$. Each respondent $i$ provides:

1. A vote $v_i \in \{A, B\}$

2. A prediction $p_i \in [0, 1]$ representing the expected fraction of respondents voting $A$

Let $\bar{v}$ denote the actual fraction voting $A$, and $\bar{p}$ denote the average prediction. The SP algorithm selects $A$ if $\bar{v} > \bar{p}$, and $B$ otherwise. In words: choose the answer that is more popular than people predicted.

**Intuition.** The algorithm exploits an asymmetry in metacognitive awareness. Consider the canonical example: "Is Philadelphia the capital of Pennsylvania?" Most people incorrectly believe yes (the majority is wrong). Those who know the correct answer (Harrisburg) also know that Philadelphia is more famous and that most people will guess wrong. When asked to predict the crowd, they forecast high support for Philadelphia. Meanwhile, those who guess Philadelphia assume their answer is common and predict similarly.

The result: nearly everyone predicts high support for Philadelphia, but the actual vote share is lower because some respondents know the truth. The "no" answer receives more votes than predicted, making it surprisingly popular. This surprise reveals the presence of informed respondents whose knowledge would be drowned out by simple majority voting.

**Worked example (Philadelphia).** Consider the question "Is Philadelphia the capital of Pennsylvania?" with answers YES and NO (truth: NO, the capital is Harrisburg). Suppose 80% of respondents are uninformed and vote YES, while 20% are informed and vote NO. Each respondent also predicts the overall fraction voting YES. Assume uninformed respondents predict $p_i = 0.95$ (they expect near-consensus on YES), while informed respondents predict $p_i = 0.80$ (they know most people will guess YES).

Let $\bar{v}_{\text{YES}}$ be the observed vote share for YES and $\bar{p}_{\text{YES}}$ the average predicted vote share for YES. Then

$$\bar{v}_{\text{YES}} = 0.80, \qquad \bar{p}_{\text{YES}} = 0.80 \cdot 0.95 + 0.20 \cdot 0.80 = 0.92.$$

SP selects YES if $\bar{v}_{\text{YES}} > \bar{p}_{\text{YES}}$, and otherwise selects NO. Here $\bar{v}_{\text{YES}} < \bar{p}_{\text{YES}}$, so SP selects NO. Equivalently, the NO answer is "surprisingly popular" because its observed share exceeds its predicted share:

$$\bar{v}_{\text{NO}} = 1 - \bar{v}_{\text{YES}} = 0.20 \quad \text{and} \quad \bar{p}_{\text{NO}} = 1 - \bar{p}_{\text{YES}} = 0.08,$$

so $\bar{v}_{\text{NO}} > \bar{p}_{\text{NO}}$. Intuitively, informed respondents both (i) vote correctly and (ii) anticipate the majority's mistake, creating a systematic prediction gap that SP exploits.

**Theoretical Guarantee.** Prelec et al. (2017) prove that under a Bayesian model where respondents share a common prior and update beliefs based on private signals, SP recovers the truth whenever the following conditions hold:

1. Respondents report their beliefs honestly

2. Respondents form unbiased predictions about others

3. There exists a subpopulation with access to the correct answer

The key insight is that truth-holders make systematically different predictions than non-truth-holders. If $A$ is correct, those who know $A$ predict lower support for $A$ than those who believe $B$ (because $A$-knowers understand that $B$-believers exist and are common). This prediction gap causes $A$ to exceed its predicted support, triggering selection by SP.

**When SP Fails.** The algorithm requires an "expert minority" structure: some respondents must know the truth while understanding that others do not. SP fails when:

- No respondent knows the answer (pure guessing)

- Everyone knows the answer (no surprise possible)

- Knowledgeable respondents cannot predict others' ignorance

- Errors are correlated such that wrong answers also exceed predictions

Our experiments test whether language models exhibit the expert-minority structure SP requires. When they do (HLE), SP yields large gains. When they do not (BoolQ), SP degrades accuracy by selecting answers that are surprisingly popular for reasons unrelated to correctness.

# B. Benchmark Construction

**Datasets, splits, and filters.** Table 1 summarizes the exact splits and filters used in our experiments.

*Table 1.* Benchmark construction and filtering used in experiments.

| Dataset | Split | Filter/Mapping | Answer | #Q |
|---|---|---|---|---|
| HLE | test | exactMatch & yes/no only | YES/NO | 35 |
| BoolQ | validation | bool → true/false | TRUE/FALSE | 100 |
| Com2Sense | validation | label → true/false | TRUE/FALSE | 100 |
| Predict-the-Future | train | normalize yes/no | YES/NO | 100 |

**Question counts and models.** We use 35 HLE questions and 100 questions each for BoolQ, Com2Sense, and Predict-the-Future.

**Sampling protocol summary.** For each (question, model) we draw 25 samples at each temperature $T \in \{0.7, 1.0\}$ for each experiment type (surprisingly popular and confidence-weighted). Each experiment issues both direct-answer prompts and prediction/confidence prompts, yielding 50 responses per (question, model, temperature, experiment). This totals 375,000 responses across all datasets.

## C. Single-Model Polling Results

Tables **??** and **??** report per-model results at $T = 1.0$. Com2Sense omits Gemma-3-4B.

## D. Prompts and Parsing Details

**Base prompts.** The prompt templates below specify the required final-answer format; for reasoning-capable models, reasoning is enabled during generation and only the final answer is parsed. We use the following templates for binary questions. For TRUE/FALSE tasks, replace YES/NO with TRUE/FALSE.

---

**Direct Answer**

```
<Question>
Answer with just YES or NO:
```

---

**Surprisingly Popular (Prediction)**

```
Question:  <Question>
What percentage of language models
will answer YES to this question?
Give just a number 0--100.
What percentage will answer NO?
(should sum to 100)
Format:  YES:XX NO:XX
```

---

**Confidence**

```
Question:  <Question>
Answer YES or NO, then rate your
confidence from 0--100.
Format:  Answer:YES Confidence:XX or
Answer:NO Confidence:XX
```

---

**Dataset-specific variants.**

**Com2Sense (Direct)**

```
<Scenario>
Does this scenario make sense?
Answer with just TRUE (makes sense)
or FALSE (doesn't make sense):
```

---

**Predict-the-Future (Direct)**

```
<Question>
You may not know the answer for
certain, but please make your best
prediction.  Answer with just YES or
NO:
```

For Com2Sense, the SP and confidence prompts use the same wording with TRUE/FALSE substitutions. For Predict-the-Future, the SP prompt adds the line "Note: The answer may not be knowable for certain, but predict based on available information."

**Parsing rules.** Direct answers are extracted by lowercasing the response and checking for the presence of the target options (YES/NO or TRUE/FALSE). If neither option is found, the response is labeled `unclear` and excluded from vote counts. Majority ties default to the second option (NO or FALSE).

For Surprisingly Popular predictions, we parse numbers using regex patterns `YES:?\s*(\d+)` (or `TRUE:?\s*(\d+)`). Missing values default to 0.5. If both option counts are zero, the observed vote rate defaults to 0.5.

For confidence responses, we parse `Answer:?\s*(yes|no|true|false)` and `Confidence:?\s*(\d+)`. Missing confidence defaults to 0.5; missing answers are treated as `unclear` and ignored in confidence-weighted voting.

# E. Temperature Stability Analysis

To test whether temperature variation induces meaningful answer diversity, we compared *plurality* answers between $T=0.7$ and $T=1.0$ for each (benchmark, model, question) tuple. Table 2 reports the fraction of cases in which the plurality answer flips.

*Table 2.* Plurality answer flip rate between $T=0.7$ and $T=1.0$. Low flip rates indicate that temperature variation does not induce the error independence required for effective aggregation.

| Benchmark | Flip Rate | Questions × Models |
|---|---|---|
| Predict-the-Future | 3.8% | 500 |
| BoolQ | 3.2% | 500 |
| HLE | 2.3% | 175 |
| Com2Sense | 1.5% | 400 |
| **Overall** | **2.9%** | 1,575 |

Flip rates vary by model: GPT-OSS-20B shows the highest instability (6.6%), while Gemma-3-4B is nearly deterministic (0.4%). GPT-OSS-20B on both BoolQ and Predict-the-Future reaches 9.0%, the highest in our evaluation. These patterns suggest smaller models may be more sensitive to temperature, though all models remain far below the diversity levels that would enable effective crowd aggregation.

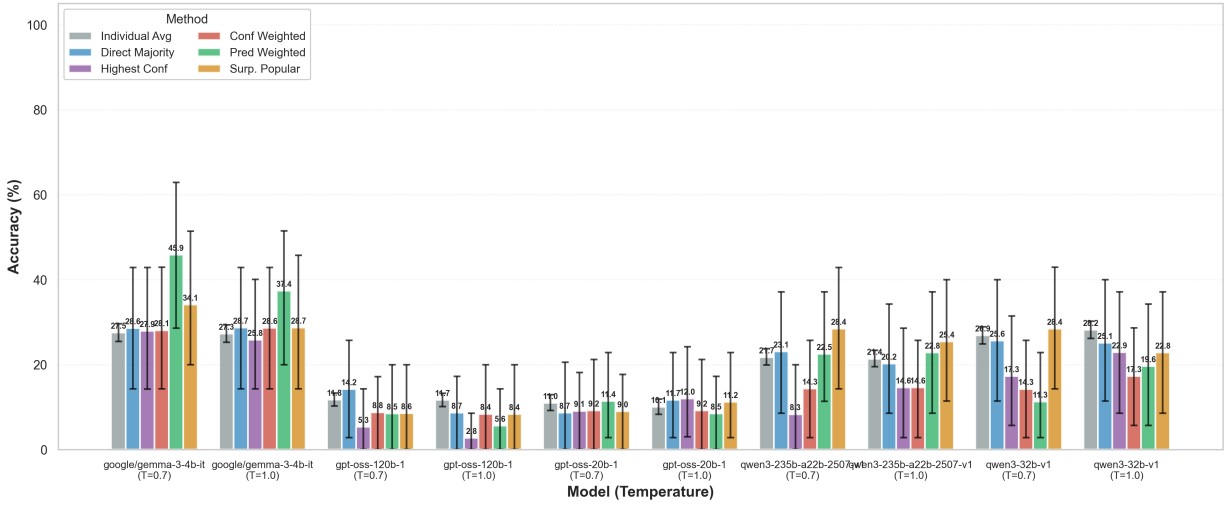

*(a)* HLE (binary subset).

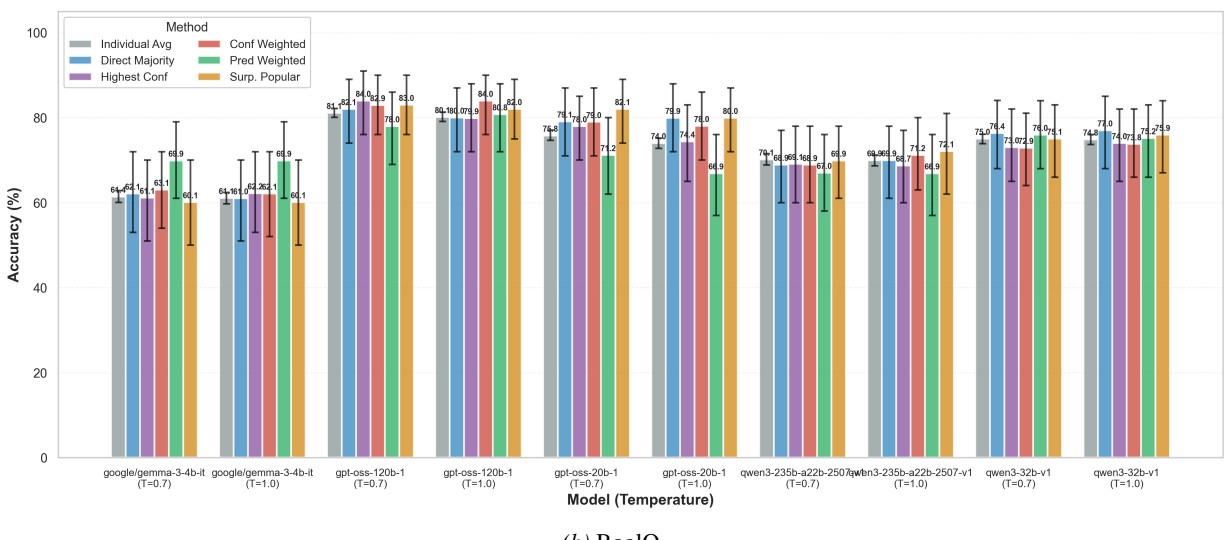

*(b)* BoolQ.

*Figure 6.* **Per-model aggregation accuracy on factual and expert benchmarks.** Accuracy by aggregation rule at temperatures $T \in \{0.7, 1.0\}$. Bars correspond to individual-sample average, direct majority vote, highest-confidence selection, confidence-weighted vote, predicted-popularity weighting, and Surprisingly Popular. Error bars show 95% bootstrap confidence intervals over questions.

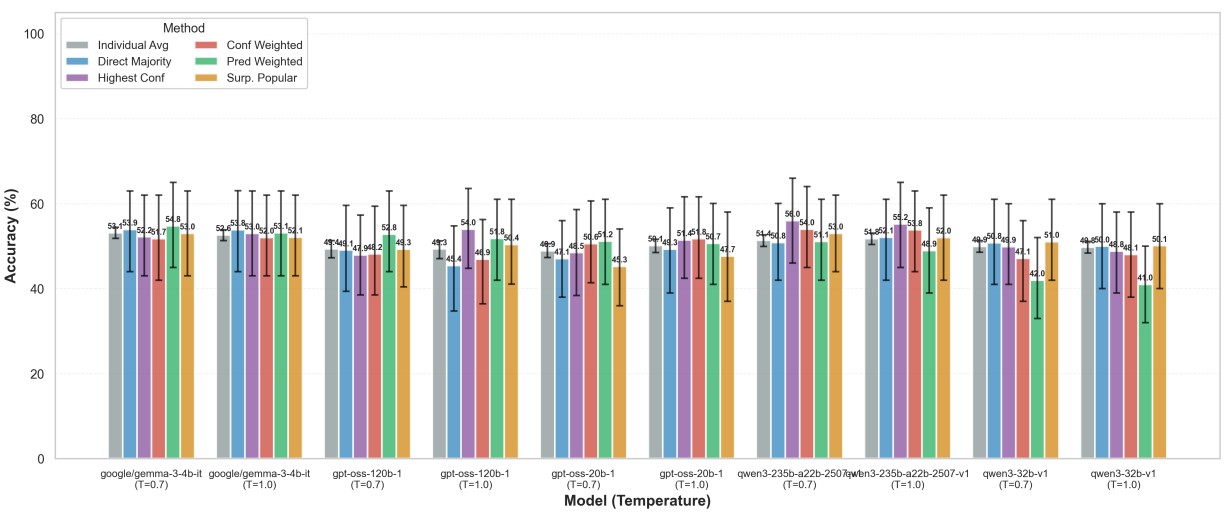

*(a)* Predict-the-Future (post-cutoff forecasting).

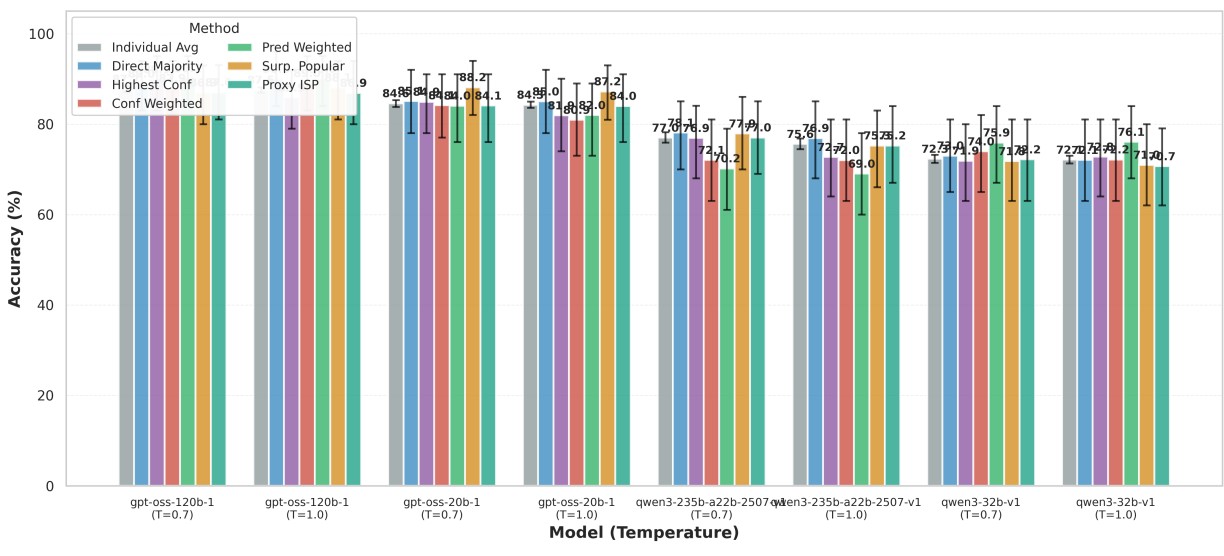

*(b)* Com2Sense.

*Figure 7.* **Aggregation fails to extract truth in verifier-absent regimes.** Forecasting and commonsense benchmarks show no consistent gains from aggregation over single-sample baselines, even as agreement increases. Error bars show 95% bootstrap confidence intervals.

