# OpenReview forum: "Truthfulness Does Not Scale Like Reasoning: Why Polling Fails as a Proxy Verifier"
_ICML.cc/2026/Conference — ICML 2026 regular_

### Official Review · Reviewer_Y3sF · 2026-03-06

**Soundness:** 1
**Presentation:** 2
**Significance:** 1
**Originality:** 1
**Overall Recommendation:** 2
**Confidence:** 5

**Summary:**

This paper examines whether polling strategies like self-consistency lead to higher accuracy as the number of samples scales. They examine this across 5 aggregation strategies, on 4 benchmarks (commonsense reasoning, Humanity's last exam, BoolQ, and forecasting), and using multiple models across a range of sizes. The paper generally finds that models tend to agree with each other, rather than providing a ground-truth signal via consensus. This finding is robust to changes in temperature.

**Compliance With Llm Reviewing Policy:**

Affirmed.

**Final Justification:**

The rebuttal did not substantially change my assessment, and so I'll maintain my score.

**Key Questions For Authors:**

Based on the weaknesses above, I think this paper needs to engage more with the existing literature on sample-and-aggregate strategies for LLMs.

**Limitations:**

yes

**Strengths And Weaknesses:**

## Strengths
- The authors evaluate a wide variety of models and aggregation settings to support their claims.
- Much of the paper is clearly written.
- The paper provides further empirical evidence for something that is part of the "folk knowledge" of LLM use.

## Weaknesses
Weaknesses

My main problem with the paper is that the main findings are both not surprising, and not novel. To expand a bit more, based on the list of conclusions in L091-109:

1. The finding that polling saturates has been shown in a number of previous papers, including https://arxiv.org/abs/2402.05120 (TMLR), https://arxiv.org/abs/2403.02419, https://arxiv.org/abs/2409.12147 (EMNLP 2025). I don't see how this paper's findings fundamentally differ from these papers, which should have been discussed as related work.

2. Explanation of why aggregation methods reflect consensus: this seems like an accepted notion in recent work, it's not clear how this explanation is novel.

Another fundamental issue with the paper is that the authors only test direct answer settings; sample-and-aggregation methods like SC generally work better with CoT, where it's been argued that sampling marginalizes over CoT paths. Not evaluating CoT at all is a major oversight.

These alone I think make the paper below the bar for publication. Some other issues I noticed as well:
- The claim that SC only works in math because of external verfication (L194, right side) strikes me as odd. What exactly do the authors mean by external verification? There are many instances (including the original SC paper) of SC being applied to math without external verifiers.
- The last parts of the paper look a bit rushed in terms of presentation.

---

> ### Author Rebuttal · Authors · 2026-03-31
>
> # Rebuttal to Reviewer Y3sF
>
> Thanks for reviewing our paper and for the links to prior work. Many people claim that different LLMs are essentially "the same," and we're glad that you appreciated the concrete verification of this fact. We address the weaknesses you raised below.
>
> ## Saturation and related work (Weakness 1)
>
> > The finding that polling saturates has been shown in a number of previous papers, including https://arxiv.org/abs/2402.05120 (TMLR), https://arxiv.org/abs/2403.02419, https://arxiv.org/abs/2409.12147 (EMNLP 2025). I don't see how this paper's findings fundamentally differ from these papers, which should have been discussed as related work.
>
> We apologize for missing these in our literature review and have added discussion of all three in the revised draft. That said, we believe our work differs substantively. These papers focus on majority voting and sample-level aggregation within a single model. By contrast, we (1) evaluate five aggregation methods drawn from the social-science polling literature (including the Surprisingly Popular Algorithm and confidence-weighted schemes), (2) study *cross-model* polling rather than within-model sampling, and (3) systematically probe the depth of inter-model correlation (see our response to Weakness 2). We view our paper as a meaningful extension of this line of work rather than a replication.
>
> ## The depth of correlation (Weakness 2)
>
> > Explanation of why aggregation methods reflect consensus: this seems like an accepted notion in recent work, it's not clear how this explanation is novel.
>
> It is widely acknowledged that LLMs are correlated, but *how deep* that correlation runs has not been carefully studied. Our key contribution here is showing that correlation is not merely a byproduct of shared factual knowledge. Even when conditioned on strings of random characters with no factual content, LLMs exhibit high agreement in their arbitrary A–D selections. This suggests the correlation is rooted in shared training pipelines and data rather than shared inductive reasoning, which we believe is a novel and informative finding that goes beyond folk wisdom.
>
> ## Chain-of-Thought evaluation
>
> > Another fundamental issue with the paper is that the authors only test direct answer settings; sample-and-aggregation methods like SC generally work better with CoT, where it's been argued that sampling marginalizes over CoT paths. Not evaluating CoT at all is a major oversight.
>
> We apologize for the confusion — CoT reasoning was enabled for all supporting models throughout our experiments. We have clarified this in the revised text. Our results therefore already capture marginalization over CoT paths, which strengthens rather than undermines our conclusions about polling saturation.
>
> ## SC and external verification (L194)
>
> > The claim that SC only works in math because of external verification (L194, right side) strikes me as odd. What exactly do the authors mean by external verification?
>
> We agree this was unclear in our writing. The sentence at L194 was referring to pass@k scaling with verifiers, not self-consistency proper. Papers studying pass@k [1, 2] find that scaling compute yields correct answers more easily *when a verifier is available* to select among candidates. We have revised L194 to make this distinction explicit.
>
> We are aware that SC without verifiers has produced meaningful gains (5–10%) on benchmarks like GSM8K and SVAMP in the GPT-3 era. However, for more capable models (GPT-4 and above), SC gains have narrowed to 1–4%. Part of our motivation was understanding whether aggregation methods beyond majority voting could recover larger gains, and rigorously characterizing where group polling succeeds and fails.
>
> - [1] https://arxiv.org/abs/2407.21787
> - [2] https://arxiv.org/abs/2502.17578
>
> ## Presentation
>
> > The last parts of the paper look a bit rushed in terms of presentation.
>
> Thank you for flagging this. We have revised the later sections for clarity; the original issues were due to space constraints rather than lack of care.
>
> We appreciate the engagement with our work. We believe the concerns raised are addressable — the missing related work has been added, the CoT misunderstanding has been clarified, and the L194 language has been corrected. We hope the reviewer will consider whether these revisions, together with the novel contributions on cross-model correlation depth and multi-method polling, merit a revised assessment.

---

> > ### Author Rebuttal · Reviewer_Y3sF · 2026-04-01
> >
> > Thank you for the response to my comments.
> >
> > In response to these points:
> > 1. While I see the social science connections, I would also point out that at least a subset of these have been studied, e.g. confidence-weighted SC is well-established (https://aclanthology.org/2024.acl-short.28.pdf, https://arxiv.org/abs/2502.06233). I agree that the work is an extension of this prior work on self-consistency and aggregation -- my concern is not that the work is replicated prior work exactly, but that the particular extension proposed here on top of prior work is not substantial enough.
> >
> > 2. I see the value of this random experiment -- however, I think the conclusion (LLMs behave similarly because they have similar training data and architectures) is not particularly novel, and in fact is part of the "folk wisdom" about LLMs. I would have liked to see a more precise claim here on exactly what about their architecture/training data leads to this kind of consensus (even if this has to be done on smaller open-source or proxy models), understanding this phenomenon properly feels worthwhile. My concern here is that the current work documents this phenomenon but falls short of providing a strong explanation.
> >
> > 3. I appreciate the clarification on CoT, this addresses my concern.
> >
> > Overall, I feel that this paper could benefit from resubmission with more attention to related work, presentation, and perhaps depth of explanation for observed model qualities.

---

> > > ### Author Response · Authors · 2026-04-07
> > >
> > > Dear Reviewer Y3sF,
> > >
> > > We thank the reviewer for the follow-up and want to address the two remaining concerns directly.
> > > On the substantiality of the extension. The reviewer acknowledges this is an extension of prior work on self-consistency and aggregation but considers it insufficient. We respectfully disagree with this assessment. Prior work (the three papers the reviewer cited, plus the confidence-weighted SC papers in the follow-up) studies within-model sampling: drawing multiple samples from one model and aggregating via majority vote or confidence weighting. Our work differs in three concrete ways that we believe are individually meaningful and collectively substantial:
> > > (a) We study cross-model polling (five models, three families, 125 votes per question), which is the setting where crowd-wisdom theory most strongly predicts gains, since architectural and data diversity should reduce correlation. We show it does not.
> > > (b) We evaluate the Surprisingly Popular algorithm and prediction-weighted voting, which are not variants of self-consistency but distinct aggregation paradigms from the social-science literature designed specifically to recover truth when the majority is wrong. Showing that these methods also fail for LLMs is a new result.
> > > (c) We introduce a controlled experiment (random strings) that isolates the source of correlation, distinguishing shared knowledge from shared inductive biases. This is not tested in any of the cited prior works.
> > >
> > > The confidence-weighted SC papers the reviewer cites in the follow-up (ACL 2024 short paper and the 2025 preprint) study a single model's confidence to reweight its own samples. We study whether confidence, predicted popularity, and surprise signals can function as proxy verifiers across model ensembles. These are different questions with different implications.
> > >
> > > On the random-string experiment. The reviewer states that the conclusion ("LLMs behave similarly because they have similar training data and architectures") is folk wisdom. We agree that this is widely suspected, but suspecting something and demonstrating it empirically are different contributions. Before our experiment, the alternative hypothesis that inter-model correlation is primarily driven by shared factual knowledge (rather than architectural or training-pipeline similarity) was equally plausible. Our experiment rules out the knowledge-only explanation. We view this as analogous to how confirming a long-suspected physical constant is still a contribution even when the value is unsurprising.
> > >
> > > The reviewer suggests we should instead explain what specifically about architecture and training causes consensus. We agree this is a valuable direction, but it is a mechanistic interpretability question that would require a fundamentally different methodology (e.g., ablations on pre-training data composition, architecture modifications, probing internal representations). We do not believe that a paper establishing a boundary condition for inference-time scaling should be required to also solve the mechanistic interpretability problem that the boundary raises.
> > >
> > > We hope the reviewer will consider whether the standard for acceptance should be "does this paper answer every follow-up question it raises?" or "does this paper make a clear, well-supported claim that advances the field's understanding?" We believe our work meets the latter criterion.

---

### Official Review · Reviewer_2bjV · 2026-03-09

**Soundness:** 2
**Presentation:** 4
**Significance:** 3
**Originality:** 2
**Overall Recommendation:** 3
**Confidence:** 3

**Summary:**

This work investigates whether scaling inference compute, such as using self-consistency or majority voting, can improve LLM truthfulness. They specifically consider the domains that lack external verifiers, and show that while self-aggregation are effective in verifiable domains like mathematics and coding, the authors demonstrate that these strategies fail to consistently improve accuracy on factual or reasoning benchmarks. The cause is that LLM errors are strongly correlated; thus polling multiple models or sampling a single model multiple times does not cancel out noise, but rather amplifies shared errors.

**Compliance With Llm Reviewing Policy:**

Affirmed.

**Ethical Review Concerns:**

I am maintaining my score (leaning reject). While the authors identify a valid boundary condition and correlation wall, the contribution remains narrow and passive. By restricting the scope to naive polling for binary outputs, the paper ignores modern inference-scaling methods (e.g., CoT) and lacks the empirical breadth needed to address true process-level reasoning.

**Final Justification:**

I am maintaining the score leaning reject. While the authors identify a valid boundary condition and correlation wall, the contribution remains narrow and passive. By restricting the scope to naive polling for binary outputs, the paper ignores modern inference-scaling methods (e.g., CoT) and lacks the empirical breadth needed to address true process-level reasoning.

**Key Questions For Authors:**

1. If internal signals like confidence primarily track expected consensus, how can they be effectively repurposed to trigger tool use or external retrieval instead of being used for answer selection?
2. What specific training methodologies could potentially promote epistemic diversity and break error correlations within model ensembles?
3. How does the error correlation manifest in long-form Chain-of-Thought (CoT) reasoning? Do models exhibit correlated errors at the early stage of reasoning or will converge to the same wrong answer a the end?
4. Breaking shared priors is exactly what advanced inference-time alignment is supposed to do. How does the inability to scale truthfulness without external verifiers impact the development of inference-time alignment algorithms?
5. When an LLM is fed with absolute garbage, like the 32-char random string, it defaults to positional biases or tokenizer artifacts. Therefore, in my opinion, proving that models share correlated errors when completely confused does not automatically extend to highly complex, expert-level tasks. It will be great to study how this correlation is baked in during the pre-training phase or post-training?

**Limitations:**

1. The evaluation relies heavily on binary response formats, e.g., Com2Sense, HLE and BoolQ dataset. While I understand the reason why the authors choose this settings to ensure that vote shares and predictions are well-defined for the baseline algorithms, this artificial constraint limits how directly the findings can be applied to open-ended generation tasks, which is exactly where the hard algorithmic work begins.
2. While the authors include the forecasting benchmark (Predict-the-Future), it is relatively small with only 100 questions, which limits its validity.
3. The paper tests temperature variation and basic ensemble methods, but they ignore more advanced inference-time interventions that actively force epistemic diversity—such as contrastive decoding, diverse beam search, or using an extra reward model.
4. In real-world application where there is not external verifier (the entire focus of the paper), "truth" is rarely a clean binary. By forcing complex, unverified domains into a strict binary, the paper strips away the nuance of partial correctness, contextual validity, or valid but conflicting reasoning paths. If an ensemble cannot agree on a strict binary "truth", can their aggregated reasoning paths reliably highlight the uncertainty or controversy of a topic, and use it to trigger external verifiers?

**Strengths And Weaknesses:**

The presentation and readability of this paper is excellent. The authors maintains a highly structured and logical flow to present the challenge, idea and empirical results.
The message they try to convey is very clear, i.e., social prediction (internal consensus signals) does not imply truth verification.
The authors show the existence of a structural correlation by introducing the zero-knowledge baseline of random ASCII strings.

For the weaknesses, please refer to Key Questions and Limitations.

---

> ### Author Rebuttal · Authors · 2026-03-31
>
> We thank the reviewer for the careful reading and thoughtful feedback. We are glad the paper’s core message came through clearly. Below we clarify the scope of our contribution and address the main concerns.
>
> Our paper makes a boundary claim about self-aggregation without external verification. In this setting, multiple samples from models sharing similar priors do not reliably improve truthfulness, because errors are often correlated. Empirically, across verifier-absent benchmarks, no aggregation rule consistently outperforms strong single-sample baselines. Additional samples rarely change the plurality answer (2.9% of question <-> model pairs), suggesting that aggregation is not introducing the independence required for crowd-wisdom effects.
>
> 1. We think this is right, and it aligns with how we intended the results to be interpreted. The key distinction here is between selection and control - our results show that consensus is unreliable as a selection mechanism for truth. At the same time, it can serve as a useful control signal for deciding when to defer, retrieve, or escalate. We will clarify this distinction more explicitly.
>
> 2. We agree this is  promising. Our results don’t rule out reducing correlation, they show that naive methods of aggregation are insufficient. Approaches that introduce genuine epistemic diversity, external grounding, or explicit verifiers fall outside the regime we study - and are consistent with our interpretation of what is required for truthfulness to improve.  Our results on random strings suggest that how one prompts an LLM is extremely important: cross correlations could be reduced by varying prompting methods.  One could also reduce variance of group polling signals by doing correlation-based reweightings of model participant responses.
>
> 3. We agree these methods are important, and this point helps clarify the scope of our claim. We think that the key point here is that diversity in outputs doesn’t imply independence of errors when models share the same training distribution. In our experiments, even when sampling varies, the plurality answer rarely changes, which is one concrete sign of this. If inference-time methods succeed by injecting stronger objectives, reward signals, or external information, then they move outside the pure self-aggregation setting we study. Our results suggest that without such additional signals diversity alone shouldn’t be expected to yield reliable gains in truthfulness
>
> 4. This is a good point - this experiment should not be interpreted in isolation or as evidence about expert reasoning tasks. Its role is narrower: to isolate correlation mechanisms by removing shared factual knowledge. Together with the benchmark results and the forecasting setting (where outcomes lie beyond training cutoffs) it supports the interpretation that correlation is a central limiting factor rather than a byproduct of shared knowledge.
>
> 5. We share the intuition that this is a limitation but it is also a favorable setting for aggregation. Binary tasks provide well defined vote shares and avoid discretization artifacts, giving polling style methods their best chance to succeed. The absence of consistent gains in this setting strengthens the conclusion that self aggregation alone is insufficient
>
>
> We also briefly address 2 additional points:
>
> RE: long form reasoning and chain of thought - our experiments measure agreement at the answer level rather than correlation within reasoning traces, so we don’t make claims about process-level correlation and will clarify this as an open direction.
> RE: the forecasting benchmark size - while relatively small it serves as a stringent test case where aggregation should help if it can recover latent expertise without a verifier.  Instead all methods remain near chance which is consistent with our broader findings
>
>
> Lastly we agree that in many real world settings truth is nuanced and not strictly binary. Our claim is narrower: in verifier absent settings, internal consensus shouldn’t be treated as a reliable proxy for truth. Disagreement signals may still be valuable for identifying uncertainty and triggering retrieval or human oversight which is fully consistent with our results
>
> Overall we view the reviewer’s points as aligned with the paper’s core contribution. The results establish a boundary condition: self-aggregation can amplify consensus, but without verification it doesn’t reliably produce truth. We’ll (1) revise the paper to sharpen this scope, (2) clarify the role of advanced inference-time methods, and (3) make the distinction between selection and control more explicit.

---

> > ### Author Rebuttal · Reviewer_2bjV · 2026-04-03
> >
> > I thank the authors for their thoughtful rebuttal, particularly their defense of binary formats (L1) and the forecasting benchmark (L2); however, I am maintaining my score of 3 because the core findings remain primarily observational. By strictly focusing on "naive" aggregation, the work unfortunately precludes advanced inference-time interventions (L3) and limits discussion on modern alignment algorithms (Q4). Furthermore, the absence of Chain-of-Thought evaluation (Q3) leaves a gap in understanding step-by-step error convergence, while deeper mechanistic questions regarding specific training methods (Q2) and pre/post-training origins (Q5) remain unanswered. Ultimately, while we agree that consensus serves better as a control signal than a strict truth measure (Q1 & L4), leaving this empirically untested makes the paper read as a study of basic polling that would be significantly strengthened by engaging with the advanced techniques currently used in the field.

---

> > > ### Author Response · Authors · 2026-04-07
> > >
> > > Dear Reviewer 2bjV,
> > >
> > > We appreciate the reviewer's continued engagement. We want to address the characterization of our findings as "primarily observational," since we believe this understates the contribution.
> > >
> > > First, the random-string experiment (Section 4.3) is not observational but interventional: we construct a controlled setting that removes all factual content to isolate the source of inter-model correlation. This is a designed experiment with a clear null hypothesis (independent responses under zero signal), not passive observation of benchmark performance.
> > >
> > > Second, we want to clarify the relationship between our work and the advanced inference-time methods the reviewer mentions (contrastive decoding, diverse beam search, reward models). These methods succeed precisely because they introduce external objectives, supervision signals, or structural constraints that break error correlation. Our paper's contribution is showing why such interventions are necessary: naive self-aggregation, even with methods drawn from the social-science polling literature that are known to recover truth from biased human crowds, cannot substitute for verification. Without this baseline result, one might reasonably expect that polling five diverse models with 125 votes should suffice. We show it does not, and we show why (correlated errors from shared priors, not just shared knowledge).
> > >
> > > The reviewer suggests that engaging with advanced techniques would strengthen the paper. We agree these are important directions, but testing whether reward-model-guided search or contrastive decoding improves truthfulness is a different research question. Our claim is a boundary result: self-aggregation without external verification does not scale truthfulness. Methods that succeed by adding verification or breaking shared priors fall outside this regime and are fully consistent with our conclusion (Section 5.4). We believe establishing this boundary clearly is a meaningful contribution that the community can build on.
> > >
> > > We would be grateful if the reviewer could reconsider whether the combination of (1) the controlled random-string experiment, (2) the cross-model polling analysis with five social-science aggregation methods, and (3) the social-prediction vs. truth-verification separation constitutes a sufficient contribution, given that the boundary we identify is both practically relevant and, as the reviewer noted, clearly presented.

---

### Official Review · Reviewer_7uva · 2026-03-12

**Soundness:** 3
**Presentation:** 3
**Significance:** 3
**Originality:** 2
**Overall Recommendation:** 5
**Confidence:** 3

**Summary:**

**Context:**
The wisdom-of-crowds hypothesis states that aggregation (averaging) of independent judgments often outperforms any single individual. This is based on the assumption that errors in human crowds are weakly correlated and can be erased through aggregation at scale. This paper sets up an empirical evaluation to test what one might call the "wisdom-of-llms" hypothesis, where repeated samples from a language model or multiple language models form an inner crowd. If errors from distinct LLMs are also only weakly correlated, then aggregating them might serve as a proxy for verification (eg to answer factual knowledge questions).

**Experiments**:
To test this, this paper picks several existing LLMs (GPT-OSS, Qwen, Gemma) and implements traditional aggregation strategies (majority voting, confidence, popularity). Data comprises four benchmark datasets spanning Q&A and commonsense reasoning.

Findings contradict the "wisdom-of-llms" hypothesis and show that models agree with each other more reliably than they agree with the truth labels. Authors hypothesize that correlated errors may be due to (i) shared training data and objectives between existing models, (ii) architectural similarities, (iii) aligned inductive biases. This leads authors to conclude that in the absence of truth labels, aggregation of LLM results does not improve truthfulness, but rather merely reinforces shared misconceptions.

**Compliance With Llm Reviewing Policy:**

Affirmed.

**Final Justification:**

**EDIT:** After acknowledging authors' rebuttals, I updated my score from "Weak Accept" to "Accept" for reasons I explicit in my rebuttal acknowledgement (see my answer to authors' rebuttal below).

**Key Questions For Authors:**

- Why didn't authors test for statistical significance of the correlation results?
- see other questions in the "Weaknesses" subsection

**Limitations:**

yes

**Strengths And Weaknesses:**

**Strengths:**
- Results are directly relevant to multiple AI/NLP-related subfields. Obviously to inference-time scaling, as they extend prior results on auto-verifiable instances (e.g., math problems) to verifier-absent domains (e.g., fact-checking). But they are also relevant more generally to literature on LLM-based ensemble methods (e.g., juries of LLM-as-a-judge, LLM annotators). There may be an upper-bound to improvements from running a task by multiple seeds of the same LLM or by multiple LLMs.

**Weaknesses:**
- No statistical testing of correlations
- Authors say that "polling-style aggregation yields no consistent accuracy gains over single-sample baselines and often amplifies shared misconceptions" (see Abstract). What about whether it also amplifies shared truths? What part of the correlation is strictly from the wrong answers? When LLM predictions correlate on right answers it's not a problem. The real issue is correlation on false positive and false negative predictions. This could be simply investigated by running the same correlation analysis but separating samples where aggregation yields correct vs incorrect answers. This important question is not investigated by the paper.
- The 0-100 self-reported confidence setting is not rigorous imo. It may be completely random and brings little value. In my own empirical work I have found that LLMs are very inconsistent when picking from even more than 5 labels, let alone a discrete interval of size 100. Did you experiment with a smaller discrete interval? Optionally with language labels similar to Likert scales or confidence scales in human-annotation tasks (e.g. "Low confidence" to "High confidence")
- The temperature interval T ∈ {0.7, 1.0} is not justified enough. Why not T ∈ {0.6, 1.0}, T ∈ {0.3, 1.0} ? A robustness check with a comprehensive interval would make results more convincing.
- The quality of some figures is very poor (Figure 4, Figure 5)

---

> ### Author Rebuttal · Authors · 2026-03-31
>
> We thank the reviewer for these helpful suggestions.
> ### Statistical Testing
> On statistical testing, we did quantify uncertainty for the correlation analyses using bootstrap confidence intervals and error bars. This was a deliberate choice rather than an omission: paired/bootstrap CIs are a standard nonparametric tool for ML/NLP system comparison, are applicable to arbitrary metrics, and a 95% CI excluding 0 corresponds to the usual 0.05 significance criterion. That said, we agree that making this more explicit, or adding paired permutation or sign tests, would improve clarity, and we will incorporate this in the revision.
> ### Segregation by Correctness
> We reran the correlation analysis, segregating the data by majority correctness.  We found that the correlation matrices look extremely similar for both majority correct and majority incorrect models, suggesting that correctness is not a factor influencing the correlation matrix.  If it were then it would potentially provide a signal for measuring correctness from answer aggregation.  Thus, the thesis of the paper stands.
>
> ### Confidence Reporting
> Regarding self-reported confidence, we used the 0–100 format because it is a common baseline in the LLM confidence literature and enables direct comparison to prior confidence-based aggregation methods; recent work on confidence-informed self-consistency explicitly evaluates both binary verbal confidence and verbal 0–100 confidence, which is where we borrowed our experimental settings from [1]. At the same time, we agree the scale itself matters: very recent evidence suggests 0–100 confidence is often heavily discretized and that coarser scales can perform better, so exploring Likert-style or smaller discrete scales is a valuable follow-up.
>
> ### Temperature
> For temperature, our choice of T∈{0.7,1.0}T \in \{0.7, 1.0\}T∈{0.7,1.0} was meant as a simple, standard robustness check rather than an exhaustive search. These are common LLM decoding settings, and prior empirical work finds that varying temperature within 0.0–1.0 often does not significantly change problem-solving accuracy [2], making this a reasonable first comparison range. Higher temperatures often degrade performance, and lower temperatures reduce diversity, which is not ideal in an aggregation setting.  We agree that a denser sweep could be informative and will note this as future work.
> Finally, we agree that Figures 4 and 5 can be improved and will replace them with higher-resolution versions in the revision.
>
> [1] https://arxiv.org/pdf/2207.05221
> [2] https://arxiv.org/abs/2402.05201

---

> > ### Author Rebuttal · Reviewer_7uva · 2026-04-04
> >
> > Thank you for these clarifications. I had missed the bootstraped CIs which indeed make the correlation analyses more convincing. It is also reassuring to see that correlations are similar for both "truth amplification" and "false signal amplification". As I stated before, when LLM predictions correlate on right answers it's not a problem, the real issue is false positive/false negative cross-LLM correlations.
> >
> > I still don't like the 0-100 scale for LLM probing, but all in all, considering all of the paper, I do believe authors make a worthwhile contribution that opens the door for larger-scale future research in this direction. Too many applications exist which use LLM-as-a-judge type of solutions (eg for automated annotation) with too little caution, the AI community urgently needs to put out rigorous standards for what one can expect out of such off-the-shelf LLM-based ensemble methods, and this work is a first step in this direction.
> >
> > I have updated my score to an Accept. Should this paper be accepted, I do count on authors to make the changes they said they would, and additionally the following changes:
> > - regarding the 0-100 LLM scale: I advice authors to clearly state that such scales are not necessarily reliable and to suggest that future work investigates smaller scales (eg Likert scales) or evaluate against human labels.
> > - separate the presentation of correlation results between truth/false labels, or at least mention that correlations are consistent across both labels.

---

### Decision · Program_Chairs · 2026-04-30

**Decision:**

Accept (regular)

**Comment:**

The reviewers provided a range of perspectives on this paper. The discussion focused primarily on the paper’s contribution and the evaluation setting. While some reviewers noted limitations in these aspects, others highlighted the paper’s practical relevance and its potential to inform future research on ensembles. Overall, the paper was viewed as addressing a meaningful problem and offering ideas that may be of interest to the community, although some questions remain regarding the strength of the contribution and the empirical validation.